

# Gynomonoecy in a mycoheterotrophic orchid *Eulophia zollingeri* with autonomous selfing hermaphroditic flowers and putatively outcrossing female flowers

Kenji Suetsugu

Department of Biology, Graduate School of Science, Kobe University, Kobe, Japan

## ABSTRACT

Most orchid species exhibit an extreme case of hermaphroditism, owing to the fusion of male and female organs into a gynostemium. Exceptions to this rule have only been reported from the subtribes Catasetinae and Satyriinae. Here, I report an additional orchidaceous example whose flowers are not always hermaphroditic. In several Japanese populations of *Eulophia zollingeri* (Rchb.f.) J.J.Sm, a widespread Asian and Oceanian orchid, some flowers possess both the anther (i.e., anther cap and pollinaria) and stigma, whereas others possess only the stigma. Therefore, pollination experiments, an investigation of floral morphology and observations of floral visitors were conducted to understand the reproductive biology of *E. zollingeri* in Miyazaki Prefecture, Japan. It was confirmed that *E. zollingeri* studied here possesses a gynomonoecious reproductive system, a sexual system in which a single plant has both female flowers and hermaphroditic flowers. In addition, hermaphroditic flowers often possess an effective self-pollination system while female flowers could avoid autogamy but suffered from severe pollinator limitation, due to a lack of agamospermy and low insect-mediated pollination. The present study represents the first documented example of gynomonoecy within Orchidaceae. Gynomonoecy in *E. zollingeri* may be maintained by the tradeoff in reproductive traits between female flowers (with low fruit set but potential outcrossing benefits) and hermaphroditic flowers (with high fruit set but inbreeding depression in selfed offspring). This mixed mating is probably important in mycoheterotrophic *E. zollingeri* because it occurs in shaded forest understorey with a paucity of pollinators.

Corresponding author
Kenji Suetsugu,
kenji.suetsugu@gmail.com

## INTRODUCTION

The Orchidaceae are one of the largest and most morphologically diverse families of land plants, and include more than 28,000 species classified into ∼760 genera (*Christenhusz & Byng, 2016*). Variations in its floral characteristics and their effects on reproductive success have long intrigued botanists since the time of Darwin (*Inoue, 1986*; *Nilsson, 1988*; *Sletvold*

& Gren, 2011). However, it is noteworthy that while the vast majority of orchid species produce only hermaphroditic flowers (i.e., flowers that possess both male and female reproductive organs; *Pannell, 2009*), a variety of sexual polymorphisms (the co-occurrence of morphologically distinct sex phenotypes within the same species) can be found in flowering plants as a whole (*Barrett, 2010*).

In fact, almost all orchid species exhibit an extreme case of hermaphroditism, owing to the fusion of male and female organs into a gynostemium (*Rudall & Bateman, 2002*). Exceptions to this rule have only been reported from the subtribes, Catasetinae and Satyriinae (*Pannell, 2009; Romero & Nelson, 1986; Huang et al., 2009*). More specifically, within Catasetinae, the members of *Catasetum* Rich. ex Kunth and *Cycnoches* Lindl. typically exhibit dioecy (i.e., unisexual individuals). In the dioecious *Catasetum*, male flowers forcibly attach a large pollinarium onto euglossine bees, and the bees subsequently avoid flowers with the same appearance (*Romero & Nelson, 1986*). Therefore, *Catasetum* populations are sexually dimorphic, and the agitated pollinators bearing their pollinia move away from the male flowers to the morphologically different female flowers (*Romero & Nelson, 1986*). In addition, within Satyriinae, *Satyrium ciliatum* Lindl. has been reported to produce both hermaphroditic and female individuals (i.e., gynodioecy; *Huang et al., 2009*). Female individuals of *S. ciliatum* can avoid pollen limitation for seed production and can be maintained in populations that experience high levels of pollen limitation, because females, via facultative parthenogenesis, can produce more seeds than do hermaphrodites (*Huang et al., 2009*). These observations helped elucidate the unusual maintenance of gender dimorphism in orchids (*Pannell, 2009*).

Here, I report an additional orchidaceous example whose flowers are not always hermaphroditic. In several Japanese populations of *Eulophia zollingeri* (Rchb.f.) J.J.Sm, a widespread Asian and Oceanian orchid, some flowers possess both the anther (i.e., anther cap and pollinaria) and stigma, whereas others possess only the stigma (Fig. 1). It is unlikely that the absence of anther cap and pollinaria is the result of removal by floral visitors, because careful dissection revealed that they were already absent before anthesis. Such a sexual system, in which plants have both female and hermaphroditic flowers co-occurring within the same plants, is called gynomonoecy. Compared with andromonoecy (male and hermaphroditic flowers within one plant) and monoecy (separate male and female flowers on the same plant), gynomonoecy remains a poorly studied sexual system, even though it occurs in 2.8–4.7% of flowering plants in at least 15 plant families (*Lu & Huang, 2006; Yampolsky & Yampolsky, 1922*).

In fact, while several hypotheses have been proposed, the adaptive significance of gynomonoecy remains largely unknown. First, the presence of the two flower types may permit flexible allocation of resources to female and male reproductive functions in response to environmental conditions (*Charnov & Bull, 1977; Lloyd, 1979*). Second, female flowers may promote outcrossing more than hermaphroditic flowers (*Marshall & Abbott, 1984*). Third, due to the lack of evidence for the above two hypotheses, it has been proposed that female flowers may boost attractiveness to pollinators (*Bertin & Kerwin, 1998*). Finally, female flowers may be favored because hermaphroditic flowers, which are more attractive in gynomonoecious plants, are more susceptible to florivory (*Bertin,*

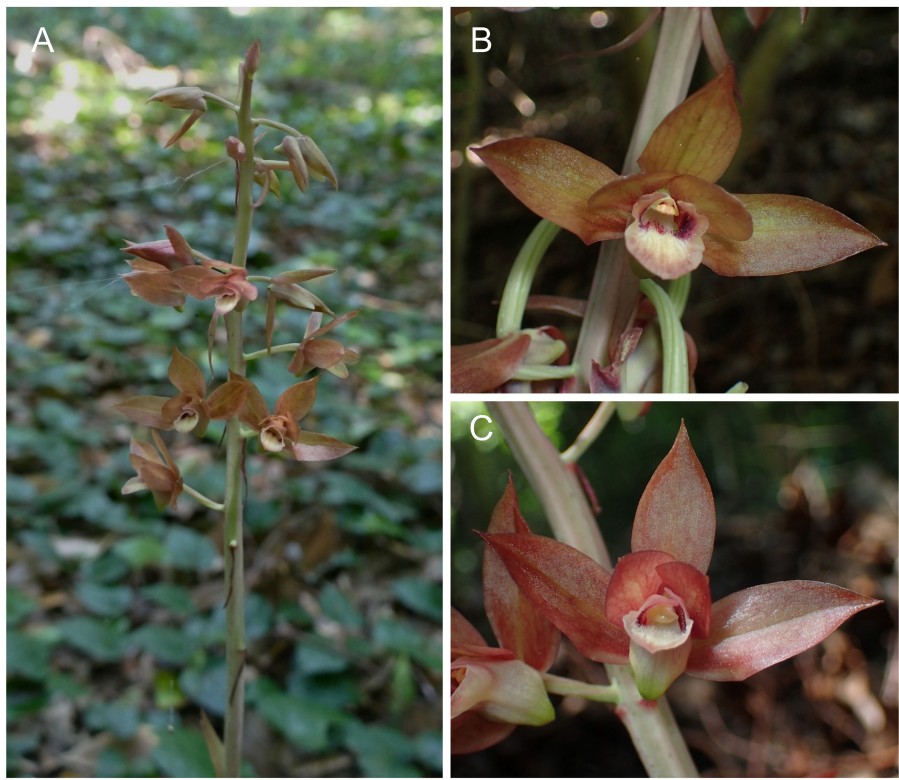

**Figure 1** ***Eulophia zollingeri* in its natural habitat.** (A) Inflorescence. (B) Hermaphroditic flower. (C) Female flower.

*Connors & Kleinman, 2010*; *Zhang, Xie & Du, 2012*). Meanwhile, current understanding of the adaptive advantages of gynomonoecy is largely limited to the Asteraceae and to a few species that have been investigated in other families (*Davis & Delph, 2005*; *Mamut et al., 2014*).

Therefore, I investigated the reproductive biology of a Japanese *E. zollingeri* population, which potentially represents the first documentation of gynomonoecy within Orchidaceae, to understand the ecological significance of the different reproductive strategies of the two floral morphs within one individual. Autonomous self-pollination has been suggested to be favorable for mycoheterotrophic plants, as they are restricted to dark shaded forest understorey with a paucity of pollinators (*Leake, 1994*; *Zhou et al., 2012*; *Suetsugu, 2013a*; *Suetsugu, 2015*). However, the Red Queen hypothesis suggests that sexual reproduction is important in a coevolutionary arms race between a parasite and host (*Ladle, 1992*), such as between mycoheterotrophic plants and fungal hosts. Intriguingly, *Zhang et al. (2014)* revealed Chinese populations of *E. zollingeri* to be hermaphroditic, non-rewarding, self-compatible, and dependent on the halictid bee *Nomia viridicinctula* Cockerell for pollination through food deception. Although those Chinese populations consequently experienced strong pollinator limitation, especially in forest understorey populations (*Zhang et al., 2014*), my preliminary investigation revealed that Japanese populations

consistently exhibit high fruit set even under the shaded forest understorey. Therefore, it is possible that the Japanese hermaphroditic flowers are capable of autonomous selfing, providing reproductive assurance, while female flowers enhance outcrossing

However, it should also be noted that the "female" *E. zollingeri* flowers can be sterile without not only the male but also the female reproductive function. In fact, in *Catasetum* species that produce male and female flowers, intermediate flowers, which are sterile, have also been found (*Romero, 1992*). Therefore, I first investigated whether the "female" flowers of *E. zollingeri* possess female reproductive functions. After that, I conducted additional pollination experiments, investigations of floral morphology, and observations of floral visitors to determine the potential for reproductive assurance provided by autonomous selfing in female flowers and outcrossing via pollinator visitation in both hermaphroditic and female flowers.

## MATERIALS & METHODS

*Eulophia zollingeri* is a mycoheterotrophic orchid distributed from India and Southeast Asia to New Guinea and Australia (*Ogura-Tsujita & Yukawa, 2008*; *Suetsugu & Mita, 2019*; *Suetsugu, Matsubayashi & Tayasu, 2020*). The behavior of floral visitors in Miyazaki City, Miyazaki Prefecture, Japan, was monitored during the peak flowering period (early to mid-July), in 2016 and 2017. Direct observations were made for ca. 30 h in total, during the peak of diurnal insect activity (09:00–17:00). The behavior of potential visitors was observed by walking around the study site, sitting next to flower patches, or hiding in the vegetation near flower clusters (within 1–2 m). In addition, artificial cross-pollination was performed in the same population in July 2016, by transferring pollinaria from different individuals to the stigmas of female flowers (five inflorescences, 10 flowers).

After confirming with the cross-pollination experiment that both hermaphroditic and female flowers produce fruits, additional pollination experiments were conducted in early July 2017. Flowers were either (i) manually cross-pollinated by transferring pollinaria to the stigmas of different individuals (five inflorescences, 10 each of female and hermaphroditic flowers); (ii) manually geitonogamous-pollinated by transferring pollinaria to the stigmas of the different flowers within the same individuals for female flowers and of the same flowers for hermaphroditic flowers (five inflorescences, 10 each of female and hermaphroditic flowers); (iii) enclosed in mesh bags to exclude floral visitors and test for autonomous self-pollination (five inflorescences, 10 hermaphroditic flowers); or (iv) left unmanipulated, in order to monitor fruit set under natural conditions (seven inflorescences, 19 female flowers and 21 hermaphroditic flowers). In addition, the relative position of female and hermaphroditic flowers on the racemes was determined in 12 inflorescences. Furthermore, to compare flower size between female and hermaphroditic flowers in gynomonoecious individuals, we measured the length of the dorsal sepal, lateral sepal, lateral petal, and lip of 10 plants using digital calipers to 0.1 mm in early July 2017. Finally, the distribution of female flowers was checked in 12 inflorescences in early July 2017. After dividing each inflorescence into distal and proximal halves, the data were tested using the Mann–Whitney $U$-test to investigate whether female flowers tended to be in the distal or basal part of the inflorescence.

**Table 1  Comparison of the length of the dorsal sepal, lateral sepal, lateral petal, and lip between hermaphroditic and female flowers in *Eulophia zollingeri*.**

| Flower type | Dorsal sepal (mm) | Lateral sepal (mm) | Lateral petal (mm) | Lip (mm) |
|---|---|---|---|---|
| Hermaphroditic | 23.5 ± 1.8 | 25.6 ± 2.1 | 17.8 ± 1.5 | 19.7 ± 0.7 |
| Female | 24.8 ± 1.6 | 26.2 ± 1.8 | 18.2 ± 1.3 | 20.0 ± 0.9 |

Notes.
  The lengths are expressed by mean ± SD (mm).

Three to four months after manual pollination, all the mature but non-dehisced fruits capsules were collected. After the fruits were silica-dried, I weighed the total mass of dry seeds freed from each capsule to the nearest 0.1 mg. All the seeds from each plant were then mixed, and 100 randomly selected seeds from each plant were examined under a dissecting microscope to determine presence of an embryo. The effects of pollination treatment on fruit set were tested using Fisher's exact test. In addition, after confirming that the datasets were normally distributed using Levene's test, the effects of pollination treatment on the seed mass, and the proportion of seeds with an embryo were tested using ANOVA, followed by Fisher's multiple comparisons test.

# RESULTS

Despite conducting ca. 30 h of field observations, few insects were observed visiting the *E. zollingeri* flowers. Several dipteran visitors, such as the agromyzid fly *Japanagromyza tokunagai*, occasionally landed on the flowers. However, none of these visitors were observed to remove or deposit pollinaria. The length of the dorsal sepal, lateral sepal, lateral petal, and lip were not significantly different between female and hermaphroditic flowers (Table 1). There are marginally significant differences in the number of female flowers between the distal half (1.2 ± 1.1; mean ± SD) and the proximal half (3.3 ± 3.4; $P = 0.06$) of the inflorescence.

More than half (6/10) of the female flowers subsequently developed fruit capsules that contained seeds with an embryo through artificial cross-pollination in 2016, thereby demonstrating their female function and confirming the gynomonoecy of the species. The results are stable at least in the investigated site, because similar results were obtained in 2017 (Table 2). The detailed pollination experiments showed that the bagged female flowers failed to develop fruits autonomously, excluding the possibility of agamospermy, while comparable fruit set ratio was also obtained in open, bagged, manual geitonogamous and allogamous hermaphroditic flowers. Therefore, the hermaphroditic flowers are capable of outbreeding, but self-compatible and not pollinator-limited for fruit set under natural condition (Table 2). The seed mass did not vary significantly with pollination treatment (ANOVA $F_{6,37} = 1.17$, $P = 0.34$), while the proportion of seeds with an embryo differed significantly among pollination treatment (ANOVA $F_{6,37} = 2.43$, $P = 0.04$). In general, the pollination experiments indicated that outcrossing tended to increase both seed mass and the number of seeds with an embryo, suggesting a negative impact of self-pollination, although the differences were not always significant (Table 2).
**Table 2  Effect of pollination treatment on fruit set, seed mass, and proportion of seeds with an embryo in *Eulophia zollingeri*.**

| Flower type | Treatment | Manual allogamy | Manual geitonogamy | Autonomous autogamy | Open |
|---|---|---|---|---|---|
| Hermaphroditic | Fruit set (%) | 70.0[a] | 70.0[a] | 50.0[a] | 57.1[a] |
| | Seed mass | 31.6 ± 11.6 | 24.1 ± 11.4 | 20.8 ± 9.3 | 24.9 ± 8.6 |
| | Seeds with an embryo | 82.4 ± 4.2[ac] | 79.0 ± 4.2[abc] | 77.2 ± 2.9[bc] | 79.2 ± 3.6[c] |
| Female | Fruit set (%) | 60.0[a] | 60.0[a] | 0.0 | 5.3[b] |
| | Seed mass | 30.2 ± 14.7 | 25.7 ± 11.0 | – | 44.9 |
| | Seeds with an embryo | 83.3 ± 4.4[a] | 77.3 ± 2.9[bc] | – | 82.0[abc] |

Notes.

Different superscript letters indicate significant differences ($P < 0.05$) between treatment groups. Both seed mass and seeds with an embryo are expressed by mean ± SD, whenever the sample size is more than > 1.

The observation of floral morphology confirmed that most of the hermaphroditic flowers possess an effective self-pollination system, in which the rostellum was poorly developed, allowing contact between the stigma and pollinaria (Fig. 2B), whereas the others had functional rostella and were therefore unlikely to be autogamous (Fig. 2D). The female flowers had a column with neither a rostellum nor anther cap and pollinaria (Fig. 2F).

## DISCUSSION

Most orchid species exhibit an extreme case of hermaphroditism, owing to the fusion of male and female organs into a gynostemium. Here I showed that a Japanese population of *Eulophia zollingeri* develops both female and hermaphroditic flowers co-occurring within the same inflorescence (i.e., gynomonoecy), while *Catasetum* and *Cycnoches* typically produces unisexual individuals (i.e., dioecy, *Romero & Nelson, 1986*), and *Satyrium ciliatum* produces both hermaphroditic and female individuals (i.e., gynodioecy; *Huang et al., 2009*). Therefore, the present study represents the first example of gynomonoecy within the Orchidaceae. However, it should be noted that gynomonoecy must not be considered a universal strategy within the species as a whole, since it was not reported in the Chinese study (*Zhang et al., 2014*). In this sense, it differs from the fixed systems in *Catasetum, Cycnoches,* and *Satyrium*. The hermaphroditic flowers of a Japanese *E. zollingeri* population often possess an effective self-pollination system, while the female flowers without agamospermy can improve the probability of outcrossing (but selfing may still occur via geitonogamous pollinations). While female flowers are generally smaller than hermaphroditic flowers in other gynomonoecious species (reviewed by *Mamut et al., 2014*), the size of floral parts did not differ significantly between female and hermaphroditic flowers of *E. zollingeri*. In addition, female flowers tend to be on the lower part of the inflorescence, suggesting that production of female flowers is not a result of resource competition. In summary, the system observed in *E. zollingeri* is consistent with the outcrossing-benefit hypothesis for gynomonoecy (*Mamut et al., 2014*).

Many models predict that plants evolve toward either complete self-fertilization or complete outcrossing (*Charlesworth & Charlesworth, 1990*). However, it seems that mixed-mating systems are more common in nature (*Vogler & Kalisz, 2001*; *Whitehead et al., 2018*),

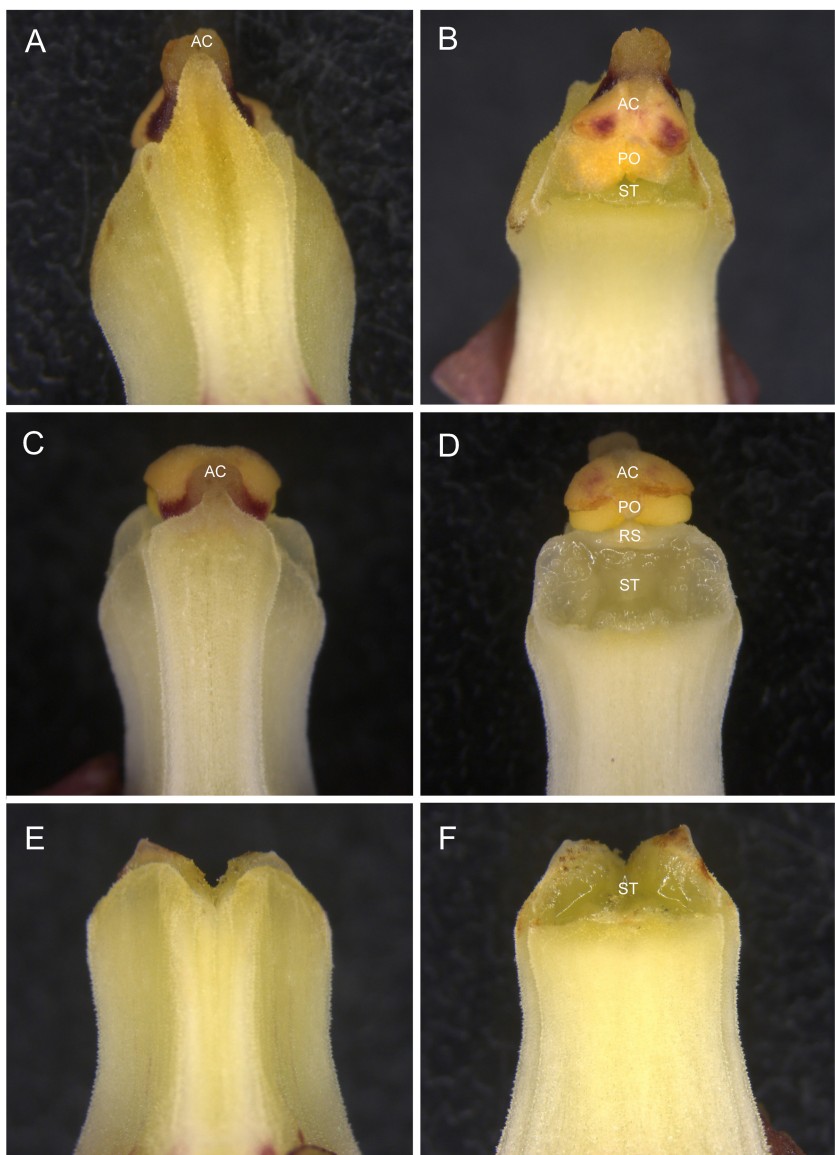

**Figure 2** **Column morphology of *Eulophia zollingeri* flowers.** (A, B) Column with a degenerate rostellum, which facilitates autogamy. (C, D) Column with a well-developed rostellum, which prevents autogamy. (E, F) Column with neither a rostellum nor anther cap and pollinaria. AC, anther cap; RS, rostellum; PO, pollinaria, ST, stigma.

possibly because mixed mating can reduce the probability of inbreeding depression via outcrossing, while still providing reproductive assurance via selfing (*Goodwillie & Weber, 2018*). As such, mixed mating systems are often referred to as "best-of-both-worlds" mating systems (*Davis & Delph, 2005*; *Goodwillie & Weber, 2018*). Mixed mating can be accomplished by delayed selfing, occurring after all other opportunities for outcrossing have been missed, because it provides reproductive assurance without limiting outcrossing opportunities. In addition, mixed mating can also occur in species that produce two flower

types within the same plant. Gynomonoecy is one of the systems involving two flower types that allows for mixed mating. Indeed, in the *E. zollingeri* populations investigated here, hermaphroditic flowers conferred reproductive assurance under pollinator-limited conditions, whereas female flowers, despite their susceptibility to pollen limitation, can facilitate outcrossing, because of the lack of autonomous selfing (Table 2).

It is possible that geitonogamy reduces the possibility of outcrossing in female flowers. In *E. zollingeri*, though, the level of geitonogamy will be low, because only a few flowers on each plant are open at one time. In particular, the risk of geitonogamy is probably negligible in nectarless *E. zollingeri*, given that pollinators are likely to quickly leave inflorescences in food-deceptive plants (*Zhang et al., 2014*; *Zhang et al., 2019*; *Suetsugu et al., 2015*). Indeed, the avoidance of geitonogamy has been hypothesized as a driving force for the evolution of food deceptive pollination in plants (*Johnson, Peter & Ågren, 2004*). Moreover, it is noteworthy that female flowers tended to be on the lower part of the inflorescence, given that *E. zollingeri* were exclusively pollinated by the halictid bee *Nomia viridicinctula* in China (*Zhang et al., 2014*) and that bees usually visit bottom flowers first and move upwards within an inflorescence (e.g., *Iwata et al., 2012*). In fact, several studies have shown that pollinator behaviors lead to directional pollen flow within inflorescences and influence floral sex allocation (*Brunet & Charlesworth, 1995*). The first flowers visited will receive more pollen grains from other plants, while the last flowers visited before pollinators leave the inflorescence tend to receive geitonogamous pollination but successfully export pollen grains to other plants (*Kudo, Maeda & Narita, 2001*). Therefore, it has been predicted that female-biased allocation to lower flowers and male-biased allocation to those in upper positions occurs in bee-pollinated plants (*Kudo, Maeda & Narita, 2001*). The variations in floral sex allocation within *E. zollingeri* are consistent with the theory and are probably effective for lowering the risk of geitonogamy.

The advantages of outcrossing and, consequently, producing female flowers can be somewhat influenced by the degree of inbreeding depression and pollinator availability (*Smithson, 2006*). In *E. zollingeri*, pollinator-mediated fruit set was arguably low, at least in the investigated population, given that (i) direct pollinator observation was unsuccessful and (ii) pollination experiments showed that natural pollination in female flowers was recorded only in one flower. Nonetheless, a small degree of outcrossing can result in a rapid decline in linkage disequilibrium across the genome and can be sufficient to overcome negative effects such as the accumulation of deleterious mutations and the slowdown in adaptation rate (*Culley & Klooster, 2007*). In addition, although the differences were not obvious (*Tremblay et al., 2005*), both artificial allogamous pollination and natural pollination in a female flower tended to increase seed mass and the proportion of seed with an embryo in *E. zollingeri* (Table 2), probably providing some support for the negative effect of autonomous selfing. It should be noted that, while seed mass and presence of an embryo was measured as the indicator of inbreeding depression, it can even under-estimate the level of inbreeding depression. Inbreeding depression might be more prominent during later stages such as seed germination or seedling growth (*Smithson, 2006*). This possibility warrants further investigation.

The outcrossing opportunity might be particularly important in mycoheterotrophic plants exploiting their mycorrhizal partners (*Suetsugu et al., 2017*), given that they usually occur in shaded understorey habitats with a paucity of pollinators, and that the Red Queen hypothesis argues that outcrossing is maintained by antagonistic interactions between a host and a parasite (*Ladle, 1992*; *Gibson & Fuentes, 2015*). Because mycoheterotrophic plants occur mainly in pollinator-hostile shaded understorey habitats, they tend to experience strong pollinator limitation, unless they possess autonomous selfing ability (*Klooster & Culley, 2009*; *Hentrich, Kaiser & Gottsberger, 2010*; *Suetsugu, 2013a*; *Suetsugu, 2015*). In fact, the Chinese populations without autogamous ability exhibited a significant difference in fruit-set between forest edge and forest populations (*Zhang et al., 2014*). Therefore, pollination limitation due to its mycoheterotrophic habit could be a driving force in the autonomous self-pollination in *E. zollingeri*. Consequently, most studies highlighted the importance of autonomous self-pollination in mycoheterotrophic plants (*Leake, 1994*; *Zhou et al., 2012*; *Suetsugu, 2013a*; *Suetsugu, 2015*). However, several recent studies have shown that mixed mating systems such as outcrossing pollinators with delayed self-pollination occur in mycoheterotrophic species belonging to Ericaceae and Gentianaceae, which evolved mycoheterotrophy independently from *E. zollingeri* (*Klooster & Culley, 2009*; *Hentrich, Kaiser & Gottsberger, 2010*). The mixed mating systems, including gynomonoecy, might be more common and important in mycoheterotrophic plants than previously thought.

Overall, it can be concluded that the Japanese population of *E. zollingeri* studied here preserve reproductive assurance by producing autonomously selfing hermaphroditic flowers and still maintain the potential benefit of producing outcrossed offspring by developing female flowers. In addition, while I did not conduct the pollination experiments, several other Japanese populations such as Okinawa ones exhibit very similar floral morphology (Fig. S1), suggesting that the strategy might be widespread at least in Japan. However, intriguingly, the Chinese populations appear to develop hermaphroditic flowers that are completely dependent on bee pollinators (*Zhang et al., 2014*). In fact, it is well-known that plant mating systems often vary widely among populations (*Suetsugu, 2013b*; *Whitehead et al., 2018*). Variations in mating systems between populations usually reflect the influence of ecological factors such as the availability and abundance of suitable pollinator (*Suetsugu, 2013b*; *Schouppe et al., 2017*; *Whitehead et al., 2018*). Therefore, it is worth clarifying how common gynomonoecy with autonomous selfing hermaphroditic flowers and putatively outcrossing female flowers, is across the distribution range and whether the strategy is be more prevalent where its effective pollinator is less abundant.

It is also notable that current understanding of the adaptive advantages of gynomonoecy is largely limited to the Asteraceae (*Marshall & Abbott, 1984*; *Bertin & Kerwin, 1998*; *Bertin, Connors & Kleinman, 2010*; *Zhang, Xie & Du, 2012*). The outcrossing hypothesis of gynomonoecy has been questioned in many asteraceous taxa, given that most Asteraceae species are self-incompatible (*Bertin & Kerwin, 1998*). However, it has been shown that hermaphroditic flowers promote seed quantity in that they are more attractive to pollinators and/or are capable of autonomous selfing, while female flowers compensate for loss of male function through outcrossing in non-asteraceous taxa [i.e., *Silene noctiflora*

(Caryophyllaceae) and *Eremurus anisopterus* (Xanthorrhoeaceae)] (*Davis & Delph, 2005*; *Mamut et al., 2014*). Taken together with these recent finding, I suggest that the ability of female flowers to reduce geitonogamy and enhance outcrossing may be widespread in gynomonoecious plants. However, it should be noted that although many orchids are (at least partially) parasitic on their mycorrhizal fungi and exhibit strong pollinator-limitation (*Leake, 1994*), gynomonoecy is not prevalent within the orchid family as a whole. Given that (i) hypotheses regarding the adaptive significance of gynomonoecy are not mutually exclusive and (ii) the seed-feeding fly *Japanagromyza tokunagai* have probably substantial negative impact on the reproduction of *E. zollingeri* (*Suetsugu & Mita, 2019*), benefits other than outcrossing, such as herbivory reduction, could also have contributed to the evolution of gynomonoecy. Therefore, further investigation is needed to elucidate the potentially diverse adaptive significance, disadvantages, and developmental constraints of gynomonoecy.

## ACKNOWLEDGEMENTS

The author thanks Drs. David Roberts, Florian Schiestl, James Ackerman and an anonymous reviewer for their constructive comments on earlier versions of the manuscript. I also thank Nobuyuki Inoue and Tadashi Minamitani for help with the field study.

### Funding
This work was financially supported by the JSPS KAKENHI Grant Number 17H05016. The funders had no role in study design, data collection and analysis, decision to publish, or preparation of the manuscript.

### Grant Disclosures
The following grant information was disclosed by the author:
JSPS KAKENHI: 17H05016.

### Competing Interests
The author declares there are no competing interests.

### Author Contributions
- Kenji Suetsugu conceived and designed the experiments, performed the experiments, analyzed the data, prepared figures and/or tables, authored or reviewed drafts of the paper, and approved the final draft.

### Data Availability
The raw measurements are available in the Supplemental Files.

### Supplemental Information
Supplemental information for this article can be found online at http://dx.doi.org/10.7717/peerj.10272#supplemental-information.

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
