# Peer review of "Gynomonoecy in a mycoheterotrophic orchid Eulophia zollingeri with autonomous selfing hermaphroditic flowers and putatively outcrossing female flowers"

_PeerJ, doi:10.7717/peerj.10272_

## Round 0.1 · original submission · Major Revisions

Many thanks for submitting your manuscript to PeerJ. As you can see all the reviewers, as well as myself, find your paper to be an intriguing study that is worthy of publication. However, the reviewers found potential issues with the interpretation of the results that need careful consideration and potentially further support. I have therefore recommended major revisions and look forward to seeing your revised manuscript and rebuttal in due course.

·

Basic reporting

Language needs minor revision

Experimental design

Very small study, design is OK

Validity of the findings

Interesting findings; I have not seen any raw data.

Additional comments

This is an interesting article documenting the presence of single sex flowers in a orchid. As most orchids are hermaphroditic, this is a worthwhile contribution to our knowledge of mating systems in orchids. I have seen a few problems that need fixing before the paper can be published.
Table 1 needs to indicate statistical values
Figure 2 needs labelling of the different organs
L122, Table 2: pollinations between different flowers of the same plant are called geitonogamy, not autogamy (i.e., an only-female flower is not capable of autogamy)
Abstract: it is mentioned that female flowers increase chance of autcrossing. If this was the case, they should be more attractive than hermaphroditic flowers, to increase the income of pollen. I guess what is really meant here is that female flowers avoid autogamy, which is semantically not the same.
There are some spacing problems in the abstract: the word possesses has a space in the middle (2 times)
Abstract: “This mixed mating” not “These”
L63: Better say “These examples” or similar, than “these studies”
L65: Better: “Here I describe”
L97: sentence starting with “however” makes no sense; reformulate.

·

Basic reporting

This manuscript is well written with only a few minor grammatical issues. The literature review is not exhaustive but is intelligently selective to get the breadth of issues to the forefront. The figures and tables are well-done and the raw data are available in the supplemental files.

L118 “After confirming gynomonoecy in E. zollingeri using the cross-pollination experiment”. Better: “After confirming with the cross-pollination experiment that both hermaphroditic and females flowers produce fruits …” It is the seed work that will confirm gynomonoecy.

L151 replace “rostella” [plural] with “rostellum” [singular]

L152 An orchid can have a functional rostellum and still be capable of self-pollination (there are various mechanisms). Replace with “… and were therefore unlikely autogamous” or words to that effect. Better yet: If all those that had a functional rostellum failed to set fruit in bagged experiments, then that is your proof that such flowers are not autogamous.

L162 Sentence should read: “… female flowers face severe pollinator …”

L162-163: The way the sentence reads, severe pollinator limitation improves chances for outcrossing. Pollinator limitation is not directly related to outcrossing probabilities. Being a female flower without agamospermy is what improves the probability of outcrossing (selfing may still occur via geitonogamous pollinations).

Experimental design

L119-127
a) It is not clear how the averages were calculated. As flowers on an inflorescence are not independent of one another (they receive resources from the same plant; pollinator attraction may be at the inflorescence level, not the flower level; etc.), the sample size for each treatment is 5, not 10. Furthermore, from an evolutionary perspective, fitness is at the plant level, not the flower.
b) ANOVAs were calculated to compare treatments for fruit and seed set. The author should state whether the data meet the assumptions of the test, and also mention which test(s) they used to determine that. Furthermore, at a minimum, F statistics and degrees of freedom must be shown, not just the P-value. This would also help the reader understand the statistical analysis.

L167: It is difficult to imagine that resource constraints would be the reason for the production of female flowers at the top (distal end) of an inflorescence because fruit production is far more expensive than male flowers. Hermaphroditic flowers may be more expensive because of pollen production, but one may argue that the female flowers would generally have genetically superior seeds and that there would be selection (if genetically based variation existed) for a favored place on the inflorescence. Unfortunately, it is not clear whether there is any fitness difference in progeny between self and outcrossed pollinations. A re-analysis of the data as suggested above is necessary.

Under resource constraints, one might expect andromonoecy to be advantageous, with male flowers produced in the upper parts of the inflorescence because they are cheap to make. Using the raw data on flower position, I divided each inflorescence into two equal halves, distal and proximal halves. Female flowers averaged 1.17 flowers in the distal half, and 3.33 flowers in the proximal half. I did not test for differences (non-parametric method would likely be appropriate) but it looks like it would be significant. So, if that is true, then there is a tendency for female flowers to be on the lower part of the inflorescence. I suggest you do this analysis or something like it.

Validity of the findings

The basic observation that Eulophia zollingerii is gynomonecious is clear. Even with the ambiguity of the current data analysis, the figures, coupled with raw and summary data, show that some hermaphroditic flowers are capable of self-pollination and this is a case of reproductive assurance. At the study site, both hermaphroditic and female flowers are rarely visited by pollinators. It is not clear whether seed set is different between self and cross pollinations. The averages are very close. Usually, if there is a difference, it is much more obvious (see table 6 in Tremblay et al. 2005, Biol J Linn Soc 84: 1-54).

Additional comments

Invoking the Red Queen hypothesis is an interesting angle and worthy of further exploration. Nonetheless, I have doubts that it could explain gynomonoecy with the parasitic orchid-fungus interaction being the underlying driver. Most evidence indicates that all orchids are parasitic on their mycorrhizal fungi, at least in early stages of development. And since orchids often have mycorrhizal fungi throughout their life, both terrestrial and epiphytic species, one might think that the Red Queen hypothesis could apply to all orchids. On the other hand, the amount of nutrients captured by orchids from their fungi may be too insignificant to register under the selection landscapes in which fungi find themselves. So, much is unknown at this point. If the Red Queen is dealing cards with orchids via their host-parasite interactions, then it should be most obvious with mycoheterotrophic species. However, it is this reviewer’s perception that most of them are autogamous – more so than the rate in which we see in autotrophic species, which if true would run contrary to expectations. You might dig into this a little more.

Reviewer 3 ·

Basic reporting

An intriguing study that is basically well conceived and carefully undertaken, but it does tend to over-interpret the data and make unnecessarily grand claims. The Results are written in a rather discursive style; the original data should be integrated more thoroughly with the text to strengthen the inferences drawn.

Individual grammatical and stylistic suggestions are given below.

Experimental design

Essentially fine to explore the ecological subject described, although the inferences are a little over-interpreted in an evolutionary context in the Discussion.

Validity of the findings

A very intriguing subject matter and clearly a rare example of gynomonoecy in the Orchidaceae. However, the Discussion tends to over-interpret the data, making rather grand conclusions from a limited – albeit intriguing – sample set. I think a more guarded approach should be adopted, particularly with using embryo presence to infer seed viability, since the two are not necessarily correlated, and in deriving support for the hypothesis that the observed mixed mating strategy is important for maintaining the species’ symbiotic (“antagonistic”) interactions. If the latter is to be viewed as a solid interpretation for the data at hand, I feel the author should consider why this reproductive strategy is not more generally prevalent within the orchid family as a whole, since many of its species are obligately mycorrhizal, many occur in pollinator-poor habitats and many exhibit pollinator-limitation. Clearly there are other important factors that also play a role.

Additional comments

Additional minor comments are listed below.

Abstract
Line 18. End first sentence with a full stop after “gynostemium”. Start second sentence with “Exceptions to this rule…”.
Line 19. Change to “Here, I report an”.
Line 23. Insert “an” before “investigation” and change following “on” to “of”.
Line 24. Insert “the” before “reproductive” and change “in” to “of”.
Line 25. Insert “a” before gynomonoecious”.
Line 28. Insert “a” before “lack”.
Line 29. Insert “documented” before “example”. Why do you claim this to be the first documented example if, as you state in the Introduction, the same phenomenon has been reported previously in Satyrium ciliatum?
Line 30. Delete “The”, and start the sentence with “Gynomonoecy…”.
Line 32. Change “These” to “This”.
Line 35. I don’t understand the meaning of “such as mycoheterotrophic plants and its fungal victims”. Change to “such as those with fungal associates”.

Introduction
Line 40. Change “is” to “are”.
Line 41. “Change “species, which are distributed among” to “species classified into”.
Line 42. Delete “The” and start sentence with “Variation in floral characteristics”.
Line 44. Insert “while” before “the vast majority”.
Line 46. Delete “while”.
Line 48. Insert “as a whole” after “flowering plants”.
Line 50. End sentence with full stop after “2002)”. Start new sentence with “Exceptions”.
Line 54. Change “dioecies” to “dioecious”.
Line 60. Change “The females of” to “Female flowers of”.
Line 65. Change “discovered” to “report”.
Line 81. Why would female flowers be more attractive to pollinators? Please explain.
Line 84. Delete “The” and start the sentence with “Current”.
Line 94. Perhaps change “The preliminary…” to “My preliminary…” otherwise it is ambiguous to which investigation you are referring.
Line 98. It is not clear what you mean here. Why do you refer to “female” flowers having male function?
Line 103. Insert “s” at end of “observation”.
Line 104. Delete “s” at end of “assurances”.
Lines 87–105. In this discussion of the mating system of Eulophia zollingeri, I am surprised you do not refer to the study of Zhang et al. (Guihaia 34: 541–547. 2014), who report data for crossing experiments and observations of pollinators in this species in China.

Materials and Methods
Line 124. Change “kept” to “enclosed”.
Line 125. Change “non-manipulated” to “left unmanipulated”
Line 132. Change “At 3–4 months” to “Three to four months”.
Line 133. Change “weighted” to “weighed”.
Line 136. By what method or measure was seed viability assessed? Since there are different assumptions involved in the different methods available (presence of an embryo, TTC-staining, in vitro germinability testing, in situ seed-baiting, etc.), you need to be more precise in describing your approach. Simple presence of an embryo should not be misconstrued as viability.

Results
Line 142. Embellishments such as “the seed-feeding fly” should be left to the Discussion, as they need a citation.
Line 145–146. Embellishments such as “It is also noteworthy…” should be avoided in the Results. Just state your findings.
Line 148. The presence of an embryo alone should not be misconstrued as viability, which would require germination experiments to verify.
Line 147. 60% is not necessarily a “high proportion”. Perhaps “More than half” is more accurate. And to which experiment are you referring here? Is this as a result of artificial self- or cross-pollination or through open pollination?
Line 150. What do you mean by “most”? Where are the data showing the results of the crossing experiments?
Lines 147–156. Please refer to Table 2 earlier in this paragraph so it obvious from what raw data you are basing these inferences.
Line 156. Change “the” to “a”.

Discussion
Line 159. To avoid confusion with the description of separate female and hermaphroditic flowers in Satyrium ciliatum that you provide in the Introduction, I think it would good to emphasise the distinguishing features of the breeding system identified here.
Line 162. Delete “a’ before “severe” and insert “s” at end of “improve”. It is not clear what the “this” refers to here. Perhaps better to restructure this sentence.
Lines 165–166. Change to ”the size of floral parts did not differ significantly in the female and hermaphroditic flowers of E. zollingeri measured in the present study.”
Line 166–167. Change to “female flowers were not only produced at the apex of the inflorescence.”
Line 169. This statement needs a citation.
Line 170. Delete “should”.
Line 180. Change “can be” to “is” or “represents”.
Lines 170–187. How does this discussion gel with the findings of Zhang et al. (Guihaia 34: 541–547. 2014), who claim E. zollingeri is food-deceptive?
Line 188. Insert “s” at end of “advantage”.
Lines 188–190. This sentence is not grammatical. Restructure.
Line 190. What is meant by “should” here? Why should it be low?
Line 193. Change to “in one female flower.”
Lines 202–205. But you have not measured actual viability in terms of germinability.
Lines 206–208. If this is the case, why is this strategy not more widely prevalent throughout the Orchidaceae, given that the vast majority of its species are obligately mycorrhizal, and that many occur in pollinator-poor habitats and exhibit pollinator-limitation? I suspect there are other factors other than those you mention here. This interpretation is a little simplistic.

---

## Round 0.2 · Minor Revisions

Many thanks for taking on many of the reviewers' comments in the revision. Like the reviewer suggest you need to be clear that you are reporting on study populations in Japan and that the results may not be the case for the species elsewhere (e.g. Zhang et al. 2014).

Reviewer 3 ·

Basic reporting

A few more typos and minor grammatical errors to be corrected as listed below. I would also suggest simplifying the title a bit, if possible.

Experimental design

Fine.

Validity of the findings

Basically as before. I feel the author should underscore the fact that what he reports for his study populations of E. zollingeri in Japan may not (in fact appears not to) be the case for the species elsewhere in its range, as evidenced by the findings of Zhang et al. (2014).

Additional comments

The title is a bit of a mouthful. Can it be simplified?

Abstract
Lines 19–20: Eulophia zollingeri is actually very widespread in Asia and Australasia, not purely a Japanese species as this line suggests. This is an important point, because the reader should be informed that the data presented here concern only a tiny fraction of the global population, and what occurs in these few Japanese populations may not reflect the situation in other parts of the species’ range. Change to “…example of Japanese populations of a widespread Asiatic orchid species whose…”.
Line 23: Add “s” at end of “observation”.
Line 31: I think you need to state what the tradeoff is in – for example, “…tradeoff in reproductive potential between female…”.
Line 33: Although I neglected to mention this in my first review of this paper, I am reticent to consider this species “fully mycoheterotrophic” – as your photos in Fig. 1 show, many populations I’ve seen of this species in Asia have green stems which probably permit a degree of nutrient gain via photosynthesis, albeit probably rather limited. In the context of the argument that you put forward in terms of gynomonoecy being differentially advantageous for such species so heavily reliant on a symbiotic partner, I would counter that, in that case, why do we not see this strategy in other truly non-photosynthetic orchids?

Introduction
Lines 95–97: I suggest rewording to “Intriguingly, Zhang et al. (2014) revealed Chinese populations of E. zollingeri to be non-rewarding, self-compatible and dependent on the Halictid bee Nomia viridicinctula Cockerell for pollination through food deception. Although those Chinese populations consequently experienced strong pollinator limitation, my preliminary investigation revealed that Japanese populations consistently exhibit high fruit set.” It is interesting to note that the Chinese study populations exhibited a significant difference in fruit-set between forest edge and forest populations (perhaps something to consider in your Discussion – e.g. were your study populations in deep forest or at the forest edge?).

Results
Line 163: Change “begged” to “bagged”.
Line 169: Change “differ” to “differed”.
Line 172: Change “while” to “although”.
Line 174–175: Delete “and lost the ability to physically separate the stigma and pollinaria”, and change “them” to “the stigma and pollinaria”.
Line 176: Insert “to be” before “autogamous”.
Line 177: Insert “a” before “column”.

Discussion
Line 182: Consistency – use “I” instead of “we”. Insert “Japanese populations of” before “Eulophia”. Change “develops” to “develop”. I think it is important to underscore that what you report in these Japanese populations of E. zollingeri is clearly not a universal strategy within the species as a whole, since it was not reported in the Chinese study (Zhang et al. 2014). In this sense, it differs from the fixed systems in Catasetum, Cycnoches and Satyrium.
Line 216: You may also wish to cite Zhang et al. (2019, BMC Plant Biology 19: 597) here.
Line 220: Insert “in China” after “viridicinctula” as you didn’t record this as the pollinator in your study populations. Does this bee species occur in Japan?
Lines 252–253: Change “system” to “systems” and “occurs” to “occur”.
Lines 254–255: Change “which have independently evolved mycoheterotrophy from E. zollingeri” to “which evolved mycoheterotrophy independently from E. zollingeri”.
Line 258: Change “can” to “might”.
Line 260: Change to “concluded that the Japanese populations of E. zollingeri studied here preserve”.
Line 271: Change “suggested” to “suggest”.
Lines 279–281: I would also add that it would be interesting to clarify how common this strategy is in different E. zollingeri populations across its range – could it be more prevalent where its effective pollinator is less abundant or absent?

Acknowledgements
Line 284: Change “Th” to “The”.

---

## Round 0.3 · accepted · Accept

Many thanks for considering the reviewers' comments. I believe the manuscript is further improved and am happy to recommend it accepted for publication.